# Mpox Surveillance and Laboratory Response in Portugal: Lessons Learned from Three Outbreak Waves (2022–2025)

**DOI:** 10.3390/idr17040086

**Published:** 2025-07-21

**Authors:** Rita Cordeiro, Rafaela Francisco, Ana Pelerito, Isabel Lopes de Carvalho, Maria Sofia Núncio

**Affiliations:** 1Emergency Response and Biopreparedness Unit, Department of Infectious Diseases, National Institute of Health Doutor Ricardo Jorge, 1649-016 Lisbon, Portugal; rafaela.godfrancisco@gmail.com (R.F.); ana.pelerito@insa.min-saude.pt (A.P.); isabel.carvalho@insa.min-saude.pt (I.L.d.C.); sofia.nuncio@insa.min-saude.pt (M.S.N.); 2Institute of Environmental Health, Faculty of Medicine, University of Lisbon, 1649-028 Lisbon, Portugal; 3Egas Moniz Interdisciplinary Research Center, Egas Moniz School of Health & Science, 2829-511 Caparica, Portugal

**Keywords:** mpox, surveillance, public health response, diagnostics, specimen types, outbreak control, Portugal

## Abstract

**Background/Objectives**: Mpox re-emerged in 2022 as a global health concern. Between 2022 and 2025, Portugal experienced three distinct outbreak waves, highlighting the critical role of laboratory surveillance and public health interventions. This study describes the epidemiological trends, diagnostic performance, and key lessons learned to improve outbreak preparedness. **Methods:** A total of 5610 clinical samples from 2802 suspected cases were analyzed at the National Institute of Health Doutor Ricardo Jorge using real-time PCR methods. Positivity rates and viral loads (Ct values) were assessed across different clinical specimen types, including lesion, anal, oropharyngeal swabs, and urine samples. **Results:** Mpox was confirmed in 1202 patients. The first outbreak accounted for 79.3% of cases (n = 953), followed by a significant reduction in transmission during subsequent waves. Lesion and rectal swabs provided the highest diagnostic sensitivity (95.1% and 87.9%, respectively). Oropharyngeal swabs contributed to diagnosis in cases without visible lesions, while urine samples showed limited utility. **Conclusions:** This study underscores the importance of sustained laboratory surveillance and adaptive public health strategies in controlling mpox outbreaks. Optimizing specimen collection enhances diagnostic accuracy, supporting early detection. Continuous monitoring, combined with targeted vaccination and effective risk communication, is essential to prevent resurgence and ensure rapid response in non-endemic regions.

## 1. Introduction

Mpox is a rare zoonotic disease caused by the monkeypox virus (MPXV), a member of the Orthopoxvirus genus within the *Poxviridae* family. MPXV was first identified in 1958 during an outbreak among monkeys in a Danish research facility. Human infection was first documented in 1970 in a nine-month-old infant in the Democratic Republic of the Congo. Endemic to West and Central Africa, mpox primarily arises in regions where frequent contact occurs between humans and wildlife or potential animal reservoirs. Transmission occurs primarily through direct contact with lesions, body fluids, or respiratory droplets from infected individuals or animals [1,2].

MPXV is currently classified into two major clades: clade I (formerly the Congo Basin clade), which includes subclades Ia and Ib, and clade II (formerly the West African clade), comprising subclades IIa and IIb. These clades differ in terms of virulence and clinical severity, with clade I being associated with more severe disease outcomes and higher mortality rates compared to clade II [1,2,3].

MPXV first gained international attention in 2003 during an outbreak in the United States. Between 2003 and 2022, Europe, particularly the United Kingdom (UK), reported only a few isolated, travel-associated cases of MPXV, primarily in individuals returning from endemic regions in Africa, namely Nigeria [1,2].

In May 2022, a global outbreak caused by clade IIb strains was identified in several non-endemic countries. Due to the rapid international spread of the virus, the World Health Organization (WHO) declared mpox a Public Health Emergency of International Concern (PHEIC) in July 2022. This decision underscored the urgent need to strengthen epidemiological surveillance and laboratory response capacities. On 14 August 2024, the WHO reissued the PHEIC declaration for mpox in response to a rise in infections caused by MPXV clade Ib in the Democratic Republic of the Congo and its spread to neighboring countries. Since then, cases of clade I mpox have been reported outside Africa, often associated with individuals with recent travel history to endemic areas. As of February 2025, the WHO had recorded over 130,000 confirmed mpox cases across 131 countries, including 304 deaths [3,4].

The typical clinical presentation of mpox includes a short febrile prodromal phase, followed by the progressive development of a characteristic skin rash with firm, umbilicated lesions. These lesions usually appear first on the head or face and subsequently spread to the limbs and trunk, maintaining a uniform stage of evolution. Lymphadenopathy is a distinguishing feature of the disease. However, in recent outbreaks atypical presentations have been reported, with lesions initially appearing in the genital area. The clinical differential diagnosis should include other exanthematous diseases, particularly varicella [1].

Laboratory diagnosis of mpox is preferably performed using nucleic acid amplification tests (NAAT), such as real-time or conventional polymerase chain reaction (PCR). These assays may be designed for general *Orthopoxvirus* detection or, preferably, specifically target MPXV. Real-time PCR is the method of choice for initial diagnostic investigation due to its high sensitivity, specificity, and fast response time. In addition, viral genome sequencing provides valuable insights into transmission dynamics and plays a crucial role in molecular surveillance efforts. The most appropriate specimen for laboratory confirmation is lesion swab, with oropharyngeal swabs also recommended to complement diagnostic sensitivity. Depending on the patient’s clinical presentation, other specimen types, such as genital and/or rectal swabs, urine, and semen, may also be considered for testing [5].

Early detection and the prompt implementation of control measures are essential to limit viral spread, underscoring the importance of reference laboratories equipped to respond rapidly to emerging health threats. In Portugal, the National Institute of Health Doutor Ricardo Jorge (INSA) has operated a dedicated Emergency Response and Biopreparedness Unit (UREB) since 2007. As the national reference laboratory for the detection of highly pathogenic agents, including *Orthopoxvirus* species, UREB plays a central role in laboratory-based surveillance, outbreak investigation, and rapid diagnostic support.

This study aims to describe the epidemiological evolution of mpox in Portugal between 2022 and 2025, characterize the major outbreak waves, and evaluate the diagnostic performance of different clinical sample types based on the experience of the national reference laboratory.

## 2. Materials and Methods

Clinical samples were collected from patients suspected of having mpox disease based on clinical observation and the national case definition, as outlined in Technical Guidance No. 004/2022 from the Portuguese Directorate-General of Health (updated 8 March 2024) [6]. According to this definition, a suspected case refers to an individual who, within 21 days prior to symptom onset, had contact with a probable or confirmed case and presented with one or more non-specific symptoms such as sudden-onset fever (≥38.5 °C), headache, myalgia, back pain, or fatigue. Alternatively, a person was also considered suspected if presenting with a sudden-onset rash (macular, papular, vesicular, or pustular), mucosal lesions, or lymphadenopathy not explained by other differential diagnoses. A probable case was defined as a person with an unexplained rash and/or anogenital complaints, together with at least one risk factor: epidemiological link to a probable or confirmed case; identifying as a man who has sex with men or a transgender person who has sex with men; history of multiple or casual partners within 21 days; positive orthopoxvirus test (excluding MPXV confirmed by PCR or sequencing); or suggestive serological evidence of orthopoxvirus infection. A confirmed case was defined as an individual with laboratory detection of MPXV DNA by real-time PCR and/or viral genome sequencing.

Sample collection was conducted across healthcare facilities nationwide in Portugal through passive universal surveillance. All specimens were submitted to the UREB-INSA reference laboratory for MPXV screening. Only samples from individuals meeting the above case definitions and with sufficient material for nucleic acid extraction were included.

According to WHO and national guidelines, when facing a suspected or probable mpox case, the following specimens should be collected: (a) one swab with exudate from a lesion or vesicular/pustular fluid placed in viral transport medium, and (b) one oropharyngeal swab in viral transport medium. The most appropriate specimen for laboratory confirmation is a lesion swab. However, oropharyngeal swabs are also recommended to improve diagnostic sensitivity. Depending on the patient’s clinical presentation, other specimen types may also be considered, such as genital and/or rectal swabs, urine, and semen [5].

Nucleic acid extraction from clinical specimens was carried out using the ANDiS Viral RNA Auto Extraction Kit on the ANDiS 350 automated platform (3DMed), following the manufacturer’s instructions. Detection of MPXV was performed using an in-house real-time PCR assay targeting the B7R gene, as previously described [7]. Each run included positive, negative, and internal controls (RNAseP) to ensure assay reliability. PCR results were interpreted based on cycle threshold (Ct) values: negative (Ct ≥ 40), weakly positive (35 ≤ Ct < 40), and positive (Ct < 35). A case was considered laboratory-confirmed when at least one sample tested positive by real-time PCR.

The positivity rate was calculated in two contexts: (1) per outbreak period, as the proportion of confirmed cases among all individuals tested; and (2) by sample type, based on the distribution of positive, negative, and indeterminate results from specimens collected from confirmed mpox-positive individuals.

For statistical analysis, sample characteristics between the positive and negative groups were compared using Fisher’s exact test, with a significance level set at *p* < 0.05, performed using GraphPad QuickCalcs (2025 GraphPad Sofware, Dotmatics, Boston, MA, USA) [8].

This study was conducted in accordance with all applicable ethical guidelines. INSA serves as the national reference laboratory in Portugal and is officially designated by the Directorate-General of Health—under Technical Orientation No. 004/2022 (31 May 2022; updated 8 March 2024)—to perform MPXV diagnostic testing and genetic characterization [6]. All samples were processed in an anonymized format, and no identifiable patient information or metadata was accessed or used.

## 3. Results

Between May 2022 and February 2025, three distinct mpox outbreak waves were identified in Portugal. The first wave, which occurred between May 2022 and March 2023, accounted for the majority of confirmed cases, with 953 cases, representing 79.3% of the total. The second wave, from June 2023 and March 2024, recorded 229 cases (19.0%), reflecting a significant decrease compared to the previous wave. The third wave began in June 2024 and recorded 20 cases (1.7%) up to 28 February 2025, totaling 1202 confirmed cases reported by INSA (Figure 1a).

During the first wave, a large volume of samples was analyzed, and a high positivity rate was observed (n = 953/2016; 47.0%), with the peak frequency recorded in June 2022. The second wave showed lower intensity, with reductions in both sample volume and positive cases (n = 229/605; 38.0%), and exhibited a more prolonged temporal distribution compared to the first. The third wave was characterized by a markedly lower positivity rate (n = 20/220; 9.0%) (Figure 1b).

Overall, the majority of cases were male (n = 1188; 98.8%), predominantly within the 20–29 (n = 375; 31.2%) and 30–39 (n = 512; 42.6%) age groups. Ages ranged from 17 to 66 years (median age of 33), with most cases occurring among men who have sex with men (MSM). Only 14 cases (1.2%) were reported in females, indicating a marked gender disparity. No cases were reported in children aged 0–9 years, and nine cases (0.7%) were reported in individuals over 60 years of age (Table 1).

Positive cases were detected in all regions of Portugal, but the highest number was reported in the Lisbon Metropolitan Area (n = 908; 75.5%), followed by the North (n = 234; 19.5%) and Centre (n = 29; 2.4%) regions (Table 1).

In the detailed analysis of each outbreak wave, the data reveal a consistent predominance of male cases and individuals in the 20–29 and 30–39 age groups, with low representation among those under 20 and adults over 60 years of age.

The Lisbon Metropolitan Area accounted for 77.9% (n = 742) of cases during the first outbreak and remained the most affected region in the second (n = 153; 66.8%) and third outbreak waves (n = 13; 65.0%), although its proportion gradually decreased. In contrast, the Northern region showed a significant increase in its share of cases, rising from 16.6% (n = 158) in the first wave to 30.1% (n = 69) in the second, and 35.0% (n = 7) in the third. Notably, unlike the previous waves, which began in the Lisbon Metropolitan Area, the third outbreak started in the Northern region. The Central, Alentejo, and Algarve regions, as well as the Autonomous Regions, maintained residual case numbers with little variation across the three outbreak waves.

A total of 5610 samples of different specimen were analyzed. Samples included lesion swabs (n = 2977, 53.1%), oropharyngeal swabs (n = 2146, 38.2%), rectal swabs (n = 287, 5.1%), urine (n = 99, 1.8%), and other specimen types (n = 101, 1.8%), such as genital, ocular, cerebrospinal fluid, semen, peripheral blood, pleural fluid, serum and bronchial secretions. The number of samples exceeds the number of patients due to multiple specimens being collected from some individuals, as well as follow-up sampling in certain clinical cases. Additionally, samples from the same patient were collected and tested across different outbreak periods. Initially, lesion and oropharyngeal swabs were prioritized, in line with WHO recommendations. As the understanding of the clinical presentation evolved, other specimen types, such as rectal swabs and urine, were progressively incorporated into the diagnostic workflow. In some cases, particularly among high-risk contacts or patients without visible skin lesions, oropharyngeal and rectal swabs were collected to support early diagnosis based on exposure risk and symptom profile.

MPXV was detected in 2075 samples from 1202 patients. To evaluate the diagnostic performance of different specimen types for MPXV detection, positivity rates were calculated as the proportion of MPXV-positive results among confirmed mpox patients tested with multiple sample types. A total of 2434 samples (including positive, negative, and indeterminate results) collected from these 1202 confirmed mpox cases were analyzed.

The highest positivity rates were observed in lesion swabs (n = 1250/1315; 95.1%) and rectal swabs (n = 109/124; 87.9%), compared to oropharyngeal swabs (n = 697/950; 73.4%) and urine samples (n = 19/45; 42.2%) (Figure 2a). A statistically significant difference in positivity rates was found among specimen types (*p* < 0.0001), except between lesion and rectal swabs.

In approximately 5% of cases during the first wave, diagnosis was achieved using oropharyngeal swabs (n = 38; 3.9%), rectal swabs (n = 5; 0.5%), or a combination of both (n = 6/953; 0.6%) in patients with negative lesion swabs. All samples were collected on the same day as the corresponding lesion swab. These individuals were tested due to known contact with MPXV-positive cases or identified risk behaviors.

Lesion and rectal swabs showed lower median Ct values (Ct = 23 and Ct = 22, respectively), indicating higher viral loads compared to oropharyngeal swabs (Ct = 31) and urine samples (Ct = 30) (Figure 2b). These findings reinforce lesion and rectal swabs as the preferred specimen types for laboratory diagnosis of mpox, due to their higher viral loads and superior sensitivity for MPXV detection. Oropharyngeal swabs remain a valuable alternative, particularly in patients without visible lesions, while urine samples may serve as complementary specimens in atypical clinical scenarios.

## 4. Discussion

This study analyzed 5610 clinical samples from probable mpox cases in Portugal between 2022 and 2025, confirming 1202 infections. Lesion and rectal swabs proved to be the most effective specimen types for MPXV detection, based on their high positivity rates and viral loads. Oropharyngeal swabs also showed diagnostic value, particularly in cases without visible lesions, enabling detection in ~5% of cases during the first outbreak [9]. Although less sensitive, urine samples offered additional diagnostic support (42.2% positivity), especially in atypical presentations. These findings highlight the importance of collecting specimens from multiple anatomical sites to enhance diagnostic accuracy. Surveillance data revealed three distinct outbreak waves, with a progressive decline in transmission intensity. Mathematical modeling identified high-risk sexual behavior as the key driver of transmission, with an estimated 120-fold higher impact in subgroups with elevated sexual activity. Vaccinated individuals were approximately eight times less likely to transmit the virus and tended to exhibit milder symptoms, underscoring the protective effect of vaccination [10]. Between June 2022 and February 2025, 21,063 vaccine doses were administered in Portugal, 93.5% as pre-exposure prophylaxis, covering 11,771 individuals [11]. This proactive approach, not widely adopted in other countries, was instrumental in mitigating transmission and may serve as a model for future outbreak preparedness and control [12].

The Portuguese outbreak followed demographic and clinical patterns similar to those reported in other countries, primarily affecting MSM and young adults. The first confirmed case in Portugal was reported on 17 May 2022, shortly after the initial detection in the UK on 6 May 2022, in a patient with travel history to Nigeria [13,14]. Retrospective data suggest the virus had been silently circulating in Portugal before official notification, which may explain the abrupt surge in early cases [9]. A mathematical model supports this hypothesis, indicating approximately 50 days of undetected MPXV circulation before detection [15]. Genomic surveillance identified only subclade IIb (B.1 lineages) throughout all three waves, with no detection of clade I or subclade Ib strains [10,16]. The third wave, originating in the Northern region, marked a substantial reduction in transmission, indicating improved epidemic control likely influenced by acquired immunity, behavioral changes, and focused public health measures.

The objective of this study was to generate robust evidence to inform mpox diagnostic strategies and outbreak response in a non-endemic setting. Strengths include centralized testing strategy, comprehensive sample volume across three outbreak waves, and real-time adaptation to emerging clinical knowledge. However, some limitations should be acknowledged: the study did not include detailed clinical histories or behavioral follow-up, and vaccination status was not uniformly available. Additionally, the selection of specimen types evolved over time, which may have introduced variability in diagnostic practices.

Despite a progressive decline in mpox cases, the continued circulation of MPXV in high-risk populations highlights the ongoing need for diagnostic preparedness, targeted surveillance, and risk-adapted public health responses. The integration of genomic monitoring with clinical, behavioral, and immunological data will be essential for anticipating and mitigating future re-emergence scenarios. Portugal’s experience, characterized by early detection, centralized testing, and proactive vaccination, offers important lessons for enhancing mpox preparedness frameworks in non-endemic settings.

## 5. Conclusions

This study provides key insights into the diagnostic performance of different specimen types and the evolving transmission dynamics of mpox in Portugal over three outbreak waves. Lesion and rectal swabs showed the highest positivity rates among the tested specimens, supporting their diagnostic utility, while oropharyngeal swabs demonstrated added value, particularly in patients without visible lesions. The progressive decline in case numbers across the three outbreak waves underscores the effectiveness of early diagnosis, targeted public health messaging, and notably, the implementation of a pre-exposure vaccination strategy. This early intervention, distinct from the approaches taken in many other countries, was a pivotal component of Portugal’s outbreak control.

## Figures and Tables

**Figure 1 idr-17-00086-f001:**
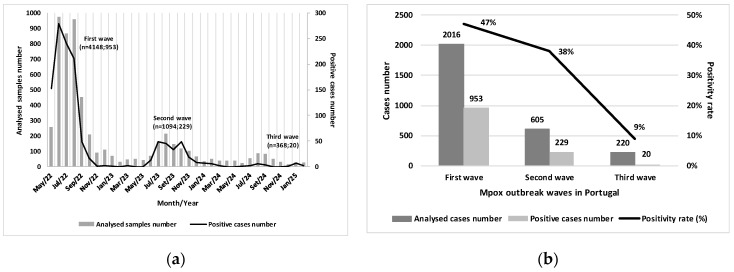
(**a**) Distribution of the number of samples analyzed and confirmed cases at INSA between 17 May 2022, and 28 February 2025; (**b**) Distribution of the number of cases analyzed, positive cases and positivity rate during the mpox outbreak wave in Portugal, between 17 May 2022, and 28 February 2025.

**Figure 2 idr-17-00086-f002:**
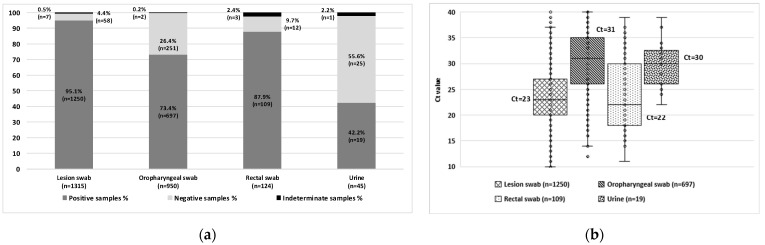
(**a**) Positivity rates across different clinical sample specimen types among confirmed cases Statistical significance was assessed using Fisher’s exact test, with a significance level set at *p* < 0.05; (**b**) MPXV viral load in positive samples, illustrated by Ct values in boxplots. Median Ct values are indicated for each specimen type.

**Table 1 idr-17-00086-t001:** Demographic data of the positive cases from the three mpox outbreak waves that occurred in Portugal, between 17 May 2022, and 28 February 2025.

Demographic Data	Positive Cases Number (%)			
	First Wave	Second Wave	Third Wave	Total
Sex				
Male	944 (99.1)	225 (98.3)	19 (95.0)	1188 (98.8)
Female	9 (0.9)	4 (1.7)	1 (5.0)	14 (1.2)
Age group				
0–9	0	0	0	0
10–19	5 (0.5)	0	0	5 (0.4)
20–29	278 (29.2)	90 (39.3)	7 (35.0)	375 (31.2)
30–39	420 (44.1)	84 (36.7)	8 (40.0)	512 (42.6)
40–49	187 (19.6)	43 (18.8)	4 (20.0)	234 (19.5)
50–59	55 (5.8)	11 (4.8)	1 (5.0)	67 (5.6)
60+	8 (0.8)	1 (0.4)	0	9 (0.7)
Portugal NUTS *				
North Region	158 (16.6)	69 (30.1)	7 (35.0)	234 (19.5)
Central Region	27 (2.8)	2 (0.9)	0	29 (2.4)
Lisbon Metropolitan Area	742 (77.9)	153 (66.8)	13 (65.0)	908 (75.5)
Alentejo Region	5 (0.5)	0	0	5 (0.4)
Algarve Region	17 (1.8)	0	0	17 (1.4)
Autonomous Region of Azores	1 (0.1)	0	0	1 (0.1)
Autonomous Region of Madeira	3 (0.3)	5 (2.2)	0	8 (0.7)
Total	953	229	20	1202

* The Nomenclature of Territorial Units for Statistics (NUTS) is developed by Eurostat, and is employed in both Portugal and the entire European Union for statistical purposes. The NUTS branch extends from NUTS I, NUTS II and NUTS III regions, with the complementary Local Administrative Units (LAU) sub-categorization being used to differentiate the local areas, of trans-national importance. NUT II categorization of Portugal regions was used.

## Data Availability

The original contributions presented in this study are included in the article. Further inquiries can be directed to the corresponding author.

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
