# Peer review of "Mpox Surveillance and Laboratory Response in Portugal: Lessons Learned from Three Outbreak Waves (2022–2025)"

_2036-7449, 2025, doi:10.3390/idr17040086_

Round 1
Reviewer 1 Report
Comments and Suggestions for Authors
- They indicate that they determined the sensitivity. However, the scope of the study was not covered in materials and methods.
- Include the periods of the epidemic waves in Portugal, as well as the diagnostic method and sample type, indicating whether they changed over time during these waves, and specify the type of surveillance (universal or sentinel). If clade Ib detection was included for 2024, specify whether the procedure included sampling at multiple times; for example, an initial time at diagnosis and another time during follow-up.
- Materials and methods. Lines 92 to 98 correspond to the results section.
- The variables to be studied (e.g., age, sex, gender, sexual preference, region, sample type, history of contact with a confirmed case, etc.) and the method of collection, sample selection criteria, or criteria for considering a case are omitted. In this regard, it is noteworthy that they mention that samples were taken from people without skin lesions and that pharyngeal swab or urine samples were collected from them. Are these samples from contacts? Therefore, specify the case definitions.
- If possible, for samples from skin lesions, specify the type of lesion.
- Detail the probes and primers used, and the method of nucleic acid extraction.
- Indicate how positivity was estimated, given that in later paragraphs they mention that indeterminate samples were included.
- Ethical considerations. Include the name of the Committee, registration number, and date of the approval report.
- Review the figures. On line 92, they report that there were 2,802 suspected cases. While in Figure 1 b, the sum of the bars is 2,841.
- On line 92, they report that there were 5,610 samples from different anatomical sites; However, on line 169, they report that there were 2,434 samples, including positive, negative, and indeterminate samples.
- The information expressed in lines 133 and 134 corresponds to the discussion.
- The information written on line 180 should not be referenced (8), as it is the results section.
- On line 202 of the discussion, they report that a clade determination was performed, which is omitted in the materials and methods section, as well as in the results section.
- Recommendation for the discussion structure. First paragraph. Essential interpretation of the main result with scientific support. Second paragraph. Compare and contrast in light of other studies; formulate hypotheses about the results, highlight unexpected results and formulate hypotheses, and indicate the strengths and limitations of the study. Third paragraph. Summarize the hypothesis and purpose of the study, your contributions to the study, and the most relevant result. Indicate unanswered questions that lead to other types of studies. In this regard, it is important to include points such as unexpected results, strengths and limitations of the study, as well as the conclusion.
- The limitations include the lack of clinical and risk history information, as well as vaccination status (the latter mentioned).
- Another lesson learned was that they implemented a vaccination strategy, which was not considered in other countries.
- Add the conclusion.
- On line 261, it says: R:C. The correct term is R.C.
- It is recommended to have the database available in a repository.
Author Response
Comment 1: They indicate that they determined the sensitivity. However, the scope of the study was not covered in materials and methods.
Response 1: Thank you for this comment. We would like to clarify that the term “diagnostic sensitivity” refers to the proportion of positive PCR results per specimen type among confirmed mpox cases. To support clarity, we added a brief explanatory sentence in the Materials and Methods section referencing how diagnostic sensitivity was calculated.
Comment 2: Include the periods of the epidemic waves in Portugal, as well as the diagnostic method and sample type, indicating whether they changed over time during these waves, and specify the type of surveillance (universal or sentinel). If clade Ib detection was included for 2024, specify whether the procedure included sampling at multiple times; for example, an initial time at diagnosis and another time during follow-up.
Response 2: Thank you for your comment. The description of the three epidemic waves was already included in the Results section, both in the main text and illustrated in Figure 1(a). The evolution in sample type selection over time was also addressed in the manuscript, initially presented in the Materials and Methods section. Following your suggestion (Comment 3), this content has now been relocated to the Results section, where it is presented in the appropriate context (lines 177 to 189). Additionally, we have now specified in the Materials and Methods that the surveillance system was passive and universal, based on nationwide case-based reporting, and not a sentinel approach (lines 95 to 96).
Comment 3: Materials and methods. Lines 92 to 98 correspond to the results section.
Response 3: Thank you for this observation. The text originally placed in lines 92 to 102 has been revised and relocated to the Results section (lines 177 to 189), where it is now appropriately integrated. The Materials and Methods section was adjusted to include only procedural and methodological descriptions.
Comment 4: The variables to be studied (e.g., age, sex, gender, sexual preference, region, sample type, history of contact with a confirmed case, etc.) and the method of collection, sample selection criteria, or criteria for considering a case are omitted. In this regard, it is noteworthy that they mention that samples were taken from people without skin lesions and that pharyngeal swab or urine samples were collected from them. Are these samples from contacts? Therefore, specify the case definitions.
Response 4: Thank you for your comment. As described in the Materials and Methods section, clinical samples were collected from individuals suspected of having mpox based on clinical observation and national case definitions (now clarified in line 94), following the guidelines issued by the Portuguese Directorate-General of Health. This included assessment of symptoms, risk exposures, and contact history. In some cases, particularly among high-risk contacts or individuals without visible skin lesions, oropharyngeal and rectal swabs were collected at the clinician’s discretion to support early diagnosis (now clarified in lines 190 to 193). We would also be happy to include the official DGS technical guidance as a reference; however, it is only available in Portuguese.
Comment 5: If possible, for samples from skin lesions, specify the type of lesion.
Response 5: Thank you for your comment. Unfortunately, detailed information on the specific type of skin lesion was not consistently available for all cases included in this study. As such, it was not possible to systematically categorize lesion types across the dataset.
Comment 6: Detail the probes and primers used, and the method of nucleic acid extraction.
Response 6: Thank you for your comment. As indicated in the Materials and Methods section, the real-time PCR protocol targeting the MPXV B7R gene was performed as previously described in the referenced study [ref. 6], which includes the full sequences of primers and probe used. Regarding nucleic acid extraction, the method and platform are also specified: extraction was carried out using the ANDiS Viral RNA Auto Extraction Kit on the ANDiS 350 automated platform (3DMed), following the manufacturer’s instructions.
Comment 7: Indicate how positivity was estimated, given that in later paragraphs they mention that indeterminate samples were included.
Response 7: Thank you for your comment. Positivity was defined as Ct < 40. Samples were classified as indeterminate when the internal control (RNaseP) failed to amplify, indicating potential sample inadequacy. These samples were included in the total number of samples analyzed per specimen type, and their frequency is presented alongside positive and negative results in Figure 1(a). Positivity rates per specimen type were calculated based on samples collected from patients with confirmed mpox, defined by the presence of at least one positive sample. For each patient, individual specimen types could be positive, negative, or indeterminate; however, the case was considered positive if any of the submitted samples met the positivity criteria. This approach is described in the Materials and Methods section, specifically in lines 113 to 114 of the revised manuscript.
Comment 8: Ethical considerations. Include the name of the Committee, registration number, and date of the approval report.
Response 8: Thank you for your comment. We would like to clarify that formal ethical approval was not required for this work. The study was conducted in full compliance with all applicable ethical guidelines. The National Institute of Health Doutor Ricardo Jorge (INSA) serves as the national reference laboratory in Portugal and is officially designated by the Directorate-General of Health to perform MPXV diagnostic testing and genetic characterization, as established under Technical Orientation No. 004/2022 (May 31, 2022; updated March 8, 2024). All samples included in the study were processed in an anonymized format. No identifiable patient data or metadata were accessed or used at any stage of the analysis. Therefore, informed consent was not applicable, and a waiver of consent was inherent to the anonymized and mandated public health nature of the work.
Comment 9: Review the figures. On line 92, they report that there were 2,802 suspected cases. While in Figure 1 b, the sum of the bars is 2,841.
Response 9: Thank you for your observation. The total number of 2,802 suspected cases, corresponding to 5,610 clinical samples, refers to the number of unique individuals tested during the study period. The difference noted in Figure 1(b) (sum of 2,841) reflects that a small number of patients were tested more than once across different outbreak periods. As such, these individuals are represented in more than one wave, contributing to the apparent discrepancy when data are presented by outbreak period.
We have now eliminated the mention of the number “2,802 suspected cases” from the Results section (line 172) to ensure consistency and avoid confusion.
Comment 10: On line 92, they report that there were 5,610 samples from different anatomical sites; However, on line 169, they report that there were 2,434 samples, including positive, negative, and indeterminate samples.
Response 10: Thank you for your comment. The total of 5,610 clinical samples refers to all specimens received and tested by the laboratory from suspected mpox cases, comprising samples from various anatomical sites. The subset of 2,434 samples mentioned in line 169 includes only the specimens collected from patients with a confirmed mpox diagnosis (i.e., those with at least one positive result).
Although multiple specimens were often collected from each individual, a single positive sample was sufficient to confirm the case. The analysis of these 2,434 samples was conducted to evaluate the diagnostic performance (positivity rate) across different sample types within the confirmed cohort, considering positive, negative, and indeterminate results per specimen type.
Comment 11: The information expressed in lines 133 and 134 corresponds to the discussion.
Response 11: Thank you for your observation. The sentence in lines 133–134 has been removed, as suggested.
Comment 12: The information written on line 180 should not be referenced (8), as it is the results section.
Response 12: Thank you for your comment. The reference to citation [8] has now been removed.
Comment 13: On line 202 of the discussion, they report that a clade determination was performed, which is omitted in the materials and methods section, as well as in the results section.
Response 13: Thank you for your comment. The clade information referred to in the Discussion corresponds to data previously published by our group and is not part of the original analyses conducted for this manuscript. To avoid confusion, the sentence has now been rephrased to clearly indicate that the findings are based on prior genomic studies, and not on new analyses presented here.
Comment 14: Recommendation for the discussion structure. First paragraph. Essential interpretation of the main result with scientific support. Second paragraph. Compare and contrast in light of other studies; formulate hypotheses about the results, highlight unexpected results and formulate hypotheses, and indicate the strengths and limitations of the study. Third paragraph. Summarize the hypothesis and purpose of the study, your contributions to the study, and the most relevant result. Indicate unanswered questions that lead to other types of studies. In this regard, it is important to include points such as unexpected results, strengths and limitations of the study, as well as the conclusion.
Response 14: Thank you for your constructive recommendation regarding the structure of the Discussion section. We appreciate your detailed guidance and agree that a clearer paragraph organization can enhance the readability and scientific coherence of the manuscript.
In response, we have restructured the Discussion into three distinct paragraphs following your suggested framework:
- The first paragraph now presents the essential interpretation of our main results, supported by relevant data and references, particularly regarding diagnostic sample performance and transmission trends.
- The second paragraph contrasts our findings with those from other studies, discusses potential explanations for the observed patterns and highlights unexpected results. We also reflect on the strengths and limitations of the study.
- The third paragraph summarizes the study's hypothesis, purpose, and key contributions, emphasizing our most relevant findings and their public health implications. We also identify unanswered questions and areas for future research.
Additionally, we have included a conclusion section, as recommended, to reinforce the key findings and public health implications of the study.
Comment 15: The limitations include the lack of clinical and risk history information, as well as vaccination status (the latter mentioned).
Response 15: Thank you for your comment. This has already been addressed in the revised Discussion section and in our response to Comment 14.
Comment 16: Another lesson learned was that they implemented a vaccination strategy, which was not considered in other countries
Response 16: Thank you for your comment. This has already been addressed in the revised Discussion section and in our response to Comment 14.
Comment 17: Add the conclusion
Response 17: Thank you for your comment. This has already been addressed in the revised Discussion section and in our response to Comment 14.
Comment 18: On line 261, it says: R:C. The correct term is R.C.
Response 18: Thank you for pointing this out. The correction has been made.
Comment 19: It is recommended to have the database available in a repository
Response 19: Thank you for your recommendation. At this stage, the dataset is not deposited in a public repository due to confidentiality considerations. However, we are open to sharing the data upon reasonable request, in line with institutional and ethical guidelines. As stated, further inquiries can be directed to the corresponding author(s).

Reviewer 2 Report
Comments and Suggestions for Authors
In the manuscript, Cordeiro and colleagues present an epidemiological study of surveillance and laboratory responses to outbreaks of monkeypox that occurred from 2022 to 2025 in Portugal. Surveillance of oral-pharyngeal swabs, lesions, and urine for Mpox in suspected cases by real-time PCR revealed that 1,202 people were positive for Mpox in the survey. The WHO has recorded over 132,797 infections, with 304 deaths occurring (a case fatality rate of ~0.22%). In Portugal, there were three outbreaks of Mpox (5/2022 to 3/2023; 6/2023 to 3/2024; and 6/2024 to 2/2025), which resulted in 953, 229, and 20 cases, respectively. These investigators used mathematical models to conclude that transmission was driven by high-risk sexual behavior. Overall, I found the manuscript to be well written, with all Figures and Tables being clear and informative. Some minor comments include the following:
1) The title should include “Portugal,” as this was a national surveillance, not an international one.
2) The authors state in the introduction that there are two major clades of Mpox, clade I (formerly the Congo basin clade) and clade II (formerly the West African clade). Did the authors determine the clades to which the Portugal isolates belonged?
Author Response
Comment 1: The title should include “Portugal,” as this was a national surveillance, not an international one.
Response 1: Thank you for this suggestion. We agree that including “Portugal” in the title helps to better reflect the scope of the study as a national surveillance effort. Accordingly, we have revised the title to: Mpox surveillance and laboratory response in Portugal: Lessons learned from three outbreak waves (2022–2025)
Comment 2: The authors state in the introduction that there are two major clades of Mpox, clade I (formerly the Congo basin clade) and clade II (formerly the West African clade). Did the authors determine the clades to which the Portugal isolates belonged?
Response 2: Thank you for your question. Yes, the clade classification of the Portuguese isolates was determined through genomic surveillance, which identified only clade IIb (B.1 sub-lineages) across all three outbreak waves in Portugal. As noted in the revised Discussion (line 239), no clade I or subclade Ib strains were detected to date. These genomic findings have been published in detail in previous studies [references 12,13].

Reviewer 3 Report
Comments and Suggestions for Authors
Cordeiro et al., summarize data of 3 human monkeypox outbrakes in Portugal, 2022-2025. The English, the figures, tables are OK. My main concern, that if not all, but at least some of the PCR products (10-20%) should have been sequenced as a final proof for specificity. The main message of the paper is, that in humans monkeypox effectively spread through rectal way, which highlight rectal swabs as the best diagnostic sample, and that homosexual male groups are mainly in risk, and there role is important in maintaining and speading the virus. It should be stated clearly in the Abtsract and discussion chapters. The authors write about these, but not clearly, and underlined enough.
The authors should take a grammar book to learn how to split words in writing.
As the rectal spread and role of homosexual groups in Monkexpox human infections are important dsicovery, I suggest the publication of this manuscript.
Introduction
The authors should have spent some sentences on history of monkeypox virus infections in Portugal and in Europe.
Math and meth.
lines 103-110 – Did the authors confirm their PCR results (at least partially) by sequencing the PCR products?
Results:
line 121 - Any information about the roots of the virus (where from it reached, was introduced to Portugal)?
line 129, 131 – 47% of what? 47% of the clinically ill persons were PCR positives? RThis 47% is not clear. What is this „positive rate”?
Figure 1. Analysed cases number. Does it mean samples from clinically suspicious persons?
lines 140-146, + Table 1. Could these epidemics be considered as an outbrake among male homosexuals? Why females are not affected? Poxviruses dominantly are not spread sexually. As I know only the tick-borne encephalitis virus is the only viral disease which shows a clear gender difference (70% male patients, everywhere). What is the reason for the high difference between the numbers of female and male Mpox patients?
Table 1. Alentejo region does not say anything to foreigners. Should be indicated as a region south of Lisbon.
line 168 and throughout the paper. - What the authors call as „patients”? Clinically ill people who are suspicious for Mpox? Or those persons whose samples proved to be PCR +?
lines 168-188. It is an important message that rectal spread is effective in cases of human Mpox infections. For further diagnostic works it should be considered. It also explaines why homosexual males were mostly infected. Probably a homosexual male Mpox infected person in acute phase of the infection arrived to Lisbon and the ourbrake erupted from homosexual communities of Lisbon. Later the virus spread sporadically to county regions. Should be concluded in the discussion.
Discussion.
line 196. Is there any information about the sexual behaviour of this first British patient?
lines 129-130 – The problem is not elavated sexual activity, but homosexuality (which means elavated sexual activity) as we know homosexuals have much frequent sexual contacts than the average heterosexual population.
Spelling:
line 39. respira-tory or respi- ratory
line 66 . abbreviations are necessary only, when the word appears frequently in the etxt.
line 74 – urine and …… comma is not necessary
line 92 – a-nalyzed, or ana-lysed
line 222 – anato-mical or ana-tomical, or anatomi-cal
line 248 – criti-cal, or cri-tical
Author Response
Comment 1: The authors should have spent some sentences on history of monkeypox virus infections in Portugal and in Europe.
Response 1: Thank you for your suggestion. A brief overview of the history of monkeypox virus infections in Europe has been included in the Introduction, specifically on lines 46-49.
Comment 2: lines 103-110 – Did the authors confirm their PCR results (at least partially) by sequencing the PCR products?
Response 2: Thank you for your question. All PCR-positive samples were further confirmed by whole genome sequencing as part of Portugal's national surveillance efforts. These genomic data have already been published in previous studies and are referenced in the Discussion section (lines 291–293). These genomic findings have been published in detail in previous studies [references 12,13].
Comment 3: line 121 - Any information about the roots of the virus (where from it reached, was introduced to Portugal)?
Response 3: Thank you for your question. Genomic analyses conducted during the early stages of the outbreak indicated that MPXV strains detected in Portugal belonged to clade IIb (lineage B.1), consistent with those identified in other countries affected during the 2022 global outbreak. These findings suggest a common origin and likely introduction via international transmission chains, as supported by published studies (Isidro et al., 2022; Borges et al., 2023) [12,13]. However, this aspect was not addressed in detail in the Results section of our manuscript, as the investigation of the phylogenetic origin of the virus falls outside the specific objectives of this study, which focused primarily on diagnostic laboratory surveillance and clinical sample performance.
Comment 4: line 129, 131 – 47% of what? 47% of the clinically ill persons were PCR positives? This 47% is not clear. What is this „positive rate”?
Response 4: Thank you for your observation. The 47% refers to the proportion of laboratory-confirmed mpox cases (by PCR) out of the total number of suspected or probable cases reported to the laboratory and for which samples were submitted. We have revised the manuscript to make this point clearer in the Results section (lines 137–141).
Comment 5: Figure 1. Analysed cases number. Does it mean samples from clinically suspicious persons?
Response 5: Thank you for your question. The “analysed samples number” shown in Figure 1 refers to the number of clinical samples tested from individuals who met the case definition for suspected mpox infection. These include suspected or probable cases reported through the national surveillance system and submitted for laboratory confirmation.
Comment 6: lines 140-146, + Table 1. Could these epidemics be considered as an outbreak among male homosexuals? Why females are not affected? Poxviruses dominantly are not spread sexually. As I know only the tick-borne encephalitis virus is the only viral disease which shows a clear gender difference (70% male patients, everywhere). What is the reason for the high difference between the numbers of female and male Mpox patients?
Response 6: Thank you for this important observation. Indeed, the mpox outbreak in Portugal, as in many other non-endemic countries during the 2022 global outbreak, was predominantly driven by transmission among men who have sex with men (MSM). This pattern has been well documented in multiple international studies [1,2], and reflects close physical contact, often sexual, as the primary mode of transmission in these settings.
Although monkeypox is not traditionally considered a sexually transmitted infection, the context of this outbreak, characterized by mucocutaneous lesions in the anogenital area and high viral loads in rectal and oropharyngeal swabs, strongly supports sexual transmission as a significant driver. The high proportion of male cases (particularly MSM) is therefore not surprising. The lower number of female cases likely reflects lower exposure within the identified transmission networks, rather than intrinsic biological resistance.
Comment 7: Table 1. Alentejo region does not say anything to foreigners. Should be indicated as a region south of Lisbon.
Response 7: Thank you for your suggestion. However, according to the NUTS (Nomenclature of Territorial Units for Statistics) classification used in Portugal and across the EU, the southern part of the country comprises more than one official region—namely, Alentejo and Algarve. For this reason, it is important to list these regions separately in Table 1 to maintain geographical accuracy and consistency with national reporting frameworks.
Comment 8: line 168 and throughout the paper. - What the authors call as „patients”? Clinically ill people who are suspicious for Mpox? Or those persons whose samples proved to be PCR +?
Response 8: Thank you for your question. In the context of this study, the term “patients” refers to individuals who were classified as probable cases based on clinical evaluation and epidemiological criteria, in accordance with the national case definition in place during the outbreak. These individuals were therefore considered clinically suspicious for mpox and had samples submitted for laboratory confirmation
Comment 9: lines 168-188. It is an important message that rectal spread is effective in cases of human Mpox infections. For further diagnostic works it should be considered. It also explaines why homosexual males were mostly infected. Probably a homosexual male Mpox infected person in acute phase of the infection arrived to Lisbon and the outbreake erupted from homosexual communities of Lisbon. Later the virus spread sporadically to county regions. Should be concluded in the discussion.
Response 9: We appreciate the reviewer’s thoughtful remarks. The data indeed support the importance of rectal swabs in diagnosing mpox, particularly in the context of the 2022–2025 outbreak in Portugal. This finding aligns with the epidemiological profile of the outbreak, which primarily affected MSM populations, as discussed in the manuscript (Discussion, lines 237-238). The potential role of rectal exposure in viral transmission and the concentration of early cases in Lisbon are consistent with other international reports and likely contributed to the outbreak's dynamics. We have ensured that these key aspects are clearly emphasized in the revised Discussion section.
Comment 10: line 196. Is there any information about the sexual behaviour of this first British patient?
Response 10: Yes, according to published reports, the first confirmed case in the UK in May 2022 had a travel history to Nigeria and was identified as part of the initial cluster involving men who have sex with men (MSM). This has been documented in reference [9] of our manuscript, which details the early epidemiological investigations conducted by UK health authorities.
Comment 11: lines 129-130 – The problem is not elevated sexual activity, but homosexuality (which means elevated sexual activity) as we know homosexuals have much frequent sexual contacts than the average heterosexual population.
Response 11: Thank you for your observation. We would like to clarify that our manuscript focuses on epidemiological and behavioral risk factors. The transmission dynamics observed in this outbreak have been associated with specific high-risk behaviors, such as having multiple partners and close physical contact, which are more prevalent in certain population subgroups, including MSM. These factors—not sexual orientation itself—are the primary contributors to the increased transmission observed in this context, as supported by the literature (e.g., references [13], [14]). We believe this framing is more appropriate from both a scientific and public health perspective.
Comment 12: The authors should take a grammar book to learn how to split words in writing/spelling.
Response 12: Thank you for your remark. We would like to clarify that the hyphenation and word splitting observed in the manuscript are automatically generated by the word processing software and not manually inserted by the authors. We will ensure that the final submitted version is carefully formatted to avoid such issues in the typeset version.

Round 2
Reviewer 1 Report
Comments and Suggestions for Authors
Materials and methods.
1) The variables to be studied and the method of collection, sample selection criteria, or criteria for considering a case are omitted.
2) Indicate how positivity was estimated.
3) They point out that the urine sample had the lowest percentage of positivity and can be used in atypical conditions. In this regard, in the Materials and Methods section, they stated that they used the operational case definitions approved in the country. If the clinical manifestations of the cases were not specified and operational case definitions were used, how did they determine that they were atypical cases? And if they were atypical, then they no longer met the operational case definition. Please indicate the operational case definitions from the surveillance system in the manuscript.
Conclusions.
4) In the conclusions, they state that the oropharyngeal swab is an alternative for those without lesions; therefore, it is reiterated that the clarification of the definitions and what constitutes an atypical case should be included. The conclusions add that they are extensive (change lines 336-343 to the discussion).
5) Finally, in the conclusions, it is necessary to modify that the samples of skin and rectal lesions are not the best or most useful; were the most positive. To determine whether they are better or have a higher viral load, a Delta Ct test is required.
Author Response
Comment 1: The variables to be studied and the method of collection,sample selection criteria, or criteria for considering a case are omitted.
Response 1: Thank you for your comment. We have revised the Materials and Methods section to explicitly include the variables studied, the method of sample collection, the sample selection criteria, and the case definition. These details have now been incorporated into the revised manuscript on lines 93–122.
Comment 2: Indicate how positivity was estimated.
Response 2: Thank you for this observation. Information on how positivity rate was estimated has now been included in the Materials and Methods section (lines 144-147), specifying the approach used for both outbreak periods and sample types.
Comment 3: They point out that the urine sample had the lowest percentage of positivity and can be used in atypical conditions. In this regard, in the Materials and Methods section, they stated that they used the operational case definitions approved in the country. If the clinical manifestations of the cases were not specified and operational case definitions were used, how did they determine that they were atypical cases? And if they were atypical, then they no longer met the operational case definition. Please indicate the operational case definitions from the surveillance system in the manuscript.
Response 3: We have revised the Materials and Methods section to explicitly include the variables studied, the method of sample collection, the sample selection criteria, and the case definition. These details have now been incorporated into the revised manuscript on lines 93–122.
Comment 4: In the conclusions, they state that the oropharyngeal swab is an alternative for those without lesions; therefore, it is reiterated that the clarification of the definitions and what constitutes an atypical case should be included. The conclusions add that they are extensive (change lines 336-343 to the discussion).
Response 4: Thank you for your comment. As suggested, the content from lines 336–343 has been moved to the Discussion section to ensure a more focused and concise Conclusion. In addition, clarification regarding the use of oropharyngeal swabs, particularly in the context of atypical presentations and cases without visible lesions, has been added to the Materials and Methods section to enhance clarity regarding sample selection and testing strategies.
Comment 5: Finally, in the conclusions, it is necessary to modify that the samples of skin and rectal lesions are not the best or most useful; were the most positive. To determine whether they are better or have a higher viral load, a Delta Ct test is required.
Response 5: Thank you for this important clarification. The conclusion has been revised to state that lesion and rectal swabs showed the highest positivity rates among the tested specimens, without implying superiority in terms of viral load or diagnostic sensitivity
